# Investment in Learning Chinese by International Students Studying Chinese as a Second Language (CSL)

**Juexuan Lu** [1], **Yajun Wang** [2], **Qi Shen** [1],*** **and Xuesong Gao** [3]

1    School of Foreign Languages, Tongji University, 1239 Siping Road, Shanghai 200092, China
2    Institute of Linguistics, Shanghai International Studies University, 550 Dalian Road (W),
     Shanghai 200083, China
3    School of Education, University of New South Wales (UNSW Sydney), Sydney, NSW 2052, Australia
*    Correspondence: qishen@tongji.edu.cn

**Abstract:** This study, drawing on the theoretical model of investment, explores what motivates and encourages international students studying Chinese as a second language (CSL) to invest in their Chinese learning using Q sorting and interview data collected from 15 international undergraduate students studying in mainland China. The results reveal that: (1) CSL students' incentives for investment are intra-personally and inter-personally diverse and can be divided into three categories (multilingual posture and cultural capital-oriented, economic capital-oriented, and cultural capital and experience-oriented); (2) CSL students' Chinese learning investment is dynamic, as they aim to enrich their learning and life experiences after studying Chinese for a period of time; (3) CSL students' investment is apparently driven by multiple perceived benefits, in that utilitarian objectives (e.g., scholarships, employment opportunities, and educational qualifications) are characteristic of CSL students' investment, but are also interwoven with some non-utilitarian objectives (e.g., enriching one's experience and making friends). The findings have some implications for CSL education and future studies.

**Keywords:** Chinese as a second language (CSL); investment; Q methodology

## 1. Introduction

In recent decades, there has been a global rise in the teaching and learning of Chinese as a second language (CSL) [1]. Though not comparable to English in terms of international status, the Chinese language, gaining momentum from China's economic power and the government's policy of promoting it, is contributing to the diversification of global linguae francae, especially against the backdrop that the COVID-19 pandemic entails multilingual capacities other than English for global mass communication [2–4]. Concomitantly, a great many studies pertaining to the teaching and learning of CSL have been conducted inside and outside Greater China [5]. Frequently explored topics in these studies range from the linguistic aspects of the Chinese language or linguistic skills (e.g., reading), learner identity, technology, or mobile-assisted teaching and learning, to some social or macro aspects of CSL such as immersive teaching and learning or language policy and planning [6–9]. Despite some attention paid to LOTE (languages other than English) teachers and learners in the Chinese context [10–13], few studies have examined the multiple incentives which motivate CSL learners' investment or the interaction between such incentives and learners' identity construction, issues which are critical to the sustainability of learning Chinese as an additional language.

Here we draw on the perspective of investment and identity ("multilingual posture") to explore how the learner's identity interacts with their Chinese learning, given that individual learners and their identities play a crucial part in the multifaceted mechanisms of language learning and teaching and language-in-education policy and planning [14–16]. Existing studies, though conducted in diverse contexts, have not profoundly delved into

the role of ideology (especially how it interacts with the milieu), and more systematic and rigorous empirical studies are necessary to enrich our understanding of CSL learners in diverse contexts.

## 2. The Investment and Identity of CSL International Students in China

Investment, as a construct and a theoretical model, is a prevalent topic in language learning studies and is frequently discussed alongside motivation. The conception of investment was initially proposed to complement the traditional psychological construct of motivation, which had been criticized for being fixed, unitary and ahistorical while ignoring the role of social structure and embedded power relations in language learning [17,18]. Despite the divergence between the theoretical foundations of the two constructs, they also share some ideas, for example identity [19].

Investment, which is based on the concepts of "capital" and "subjectivity" [20,21], is a process in which learners spend time and effort learning their target language to "make a good return" (i.e., to obtain a spectrum of material or symbolic capital and thus increase the value of their cultural capital); their investment in the target language is also an investment in their multiple, dynamic social identities [17]. Later, a tripartite model of investment was developed which regards language learners' investment as the interplay of *identity* (multiple and constantly changing; a site of struggle), *ideology* (dominant ways of thinking shaped by in-context power relations), and *capital* (either in economic, cultural, or social forms) [22]. According to this theoretical model, learner identity (positioning) is always co-determined by capital (including *affordances*, what learners own, and *perceived benefits*, what learners want) and social ideologies (the in-context valuing mechanisms that confirm or deny the value of learners' *affordances* and their access to *perceived benefits*). In this sense, learners' perceived benefits fall into the category of capital, whether in its economic, cultural, or social form.

On the other hand, the framework of motivation, drawing from the perspective of investment, has been updated by taking into account the aspect of self-identity construction pertaining to language learning, e.g., the L2 motivational self system [23,24]. Moreover, in contrast to *integrativeness*, which is linked to a specific L2 community [25], it is contended that learner identity (especially their *ideal self* in the L2 motivational self system) can develop from *international posture*, a tendency to self-identify as a member of the international community [26], or *multilingual posture* on this basis, i.e., an inclination to self-identify as a member of an international community mediated by multilingualism [27,28]. Here, the concept of *posture* echoes *imagined identities*, a term which refers to learners' self-identification with an imagined community (e.g., an international/multilingual community) [22,29]. However, the two concepts also come with theoretical differences: "posture" describes a kind of tendency or inclination pertinent to individual learners' target language(s), but "imagined identities", e.g., "transcultural identity", describes the exercise of individual agency in language learning/teaching by mediating two distinct cultures in language teaching/learning practices [30].

Despite the wide use of the model of investment in the context of English learning [31–34], including in cases of Chinese English learners with various backgrounds [35–37], more evidence is needed to confirm its interpretive power in the context of Chinese (as one of the languages other than English) learning and teaching due to the following factors: (1) the interference of English in Chinese learning; (2) the contrast between the global community of English speakers and the specific, regional communities of Chinese speakers; (3) the imbalance between the substantial social support for English learning and the relatively marginal support for Chinese learning; (4) the specificity of Chinese learning goals; and (5) the possibility of unconscious resistance to Chinese learning resulting from the worldwide superiority of English [38]. Factor (1) may serve as a kind of systemic pattern of control of CSL learner investment, and Factors (2) and (3) may indicate the possibility of a lack of perceived benefits and affordances, respectively. These factors necessitate more

studies on CSL education in order to enrich our understanding thereof and facilitate the learning of CSL by international students in China.

To fill the research gaps, this study draws on the model of investment [22] and the Q methodology [39] to answer the following questions:

(1) What kinds of perceived benefits motivate international students to invest in Chinese learning?
(2) What are the factors which influence or form the benefits perceived by international students?

### 3. The Study

#### 3.1. Research Design

We chose a university in Shanghai that has an excellent reputation for foreign language education and research and has established a series of scholarships funded by the national and municipal governments, as well as the university itself, to attract international students, including the CSC (scholarship provided by the Chinese Scholarship Council), the CIS (Confucius Institute Scholarship), the SGS (Shanghai Government Scholarship), the university's own scholarship, and the SCSE scholarship ("Outstanding Students Award" and "Diligent Students Award", offered by the university). According to its official website, it offers courses in 46 languages, and the number of international students enrolled annually in recent years has reached around 4000. We considered this university to be quite representative given the large population of international students.

Data collection was conducted through a combination of Q methodology [39] and in-depth interviews, and the analysis was based on the investment model [17,22]. The Q methodology was applied because it combines the advantages of both qualitative and quantitative methods: it not only gathers a more exhaustive collection of data concerning informants' beliefs, attitudes, and opinions compared with questionnaire surveys, but also applies factor analysis through Q analysis, which guarantees the verifiability of Q sorted data compared with pure interview data (which are usually criticized for their "subjectivity").

#### 3.2. Participants

All the participants in this study were international students enrolled in the Teaching Chinese to Speakers of Other Languages (TCSOL) program, the educational objective of which is to equip the students with not only knowledge of Chinese culture and literature and Sino–foreign cultural exchanges, but also the expertise they need to work in government, business, public institutions, educational institutions, and the media, both in and outside China. The TCSOL program in this university was established in 1984 (making it one of the earliest established TCSOL programs in China), and the annual number of international students admitted remains stable at around 30. Hence, we believe that these international students are representative of their counterparts in China.

There were 113 international students in this program during this study (excluding first-year students, whose programs are not determined until the completion of the first year of study). Their home countries and cultural backgrounds were diverse. During the first two academic years, they took EMI (English as the medium of instruction) courses which taught basic Chinese language skills, and in the third and fourth years they attended more specialized CMI (Chinese as the medium of instruction) courses related to a wide range of disciplines such as Chinese linguistics, CSL pedagogy, cross-/intercultural communication, Chinese culture, Chinese literature, and so on.

It should be noted that their HSK (*Hanyu Shuiping Kaoshi*, the Chinese proficiency test) scores were closely related to their program of study. First, applicants should have a minimum score of 180 in HSK 4 to gain admission. Second, international students can obtain a scholarship (either full or partial) sponsored by the Confucius Institute in the first year of study as long as they have a minimum score of 210 in HSK 4 and a minimum score of 60 in intermediate HSKK (*Hanyu Shuiping Kouyu Kaoshi*, the test for spoken Chinese). Since such scholarships are granted on an annual basis, awards for the subsequent years are subject to the students' academic performance. Third, HSK scores also played a gatekeeping

role when the students were going to the higher grade or fulfilling their program of study: first-year students need a minimum score of 210 in HSK 4 before continuing their second year of study, and the minimum scores for third- and fourth-year study are 205 in HSK 5 and 180 in HSK 6, respectively; they should also achieve a score of at least 205 in HSK 6 to obtain their degree. Some basic information on the participants is presented in Table 1.

**Table 1.** Profiles of the participant group (P set).

| Participants | Gender | Nationality | Funding Types | Chinese Proficiency |
|---|---|---|---|---|
| ME | F | Bangladesh | Confucius Institute Scholarship | HSK 6 |
| HU | F | Thailand | Confucius Institute Scholarship | HSK 5 |
| BE | F | Pakistan | Confucius Institute Scholarship | HSK 6 |
| BO | M | Azerbaijan | Confucius Institute Scholarship | HSK 6 |
| LI | F | Thailand | Confucius Institute Scholarship | HSK 6 |
| JI | F | Russia | Chinese Government Scholarship | HSK 6 |
| GU | M | Cambodia | Confucius Institute Scholarship | HSK 6 |
| MU | M | Morocco | Confucius Institute Scholarship | HSK 6 |
| NA | F | Columbia | Self-funded | HSK 6 |
| JIN | M | Republic of Korea | Self-funded | HSK 6 |
| YA | F | Russia | Self-funded | HSK 5 |
| HE | M | Republic of Korea | Confucius Institute Scholarship | HSK 6 |
| RU | F | Nigeria | Confucius Institute Scholarship | HSK 5 |
| YI | M | Pakistan | Confucius Institute Scholarship | HSK 6 |
| WE | F | Pakistan | Confucius Institute Scholarship | HSK 6 |
| AI | M | Tajikistan | Chinese Government Scholarship | HSK 6 |
| YIN | F | Portugal | Self-funded | HSK 5 |
| MEN | F | Pakistan | Confucius Institute Scholarship | HSK 5 |
| WA | M | Nepal | Confucius Institute Scholarship | HSK 6 |

Before conducting the Q methodology, we gathered as many statements pertaining to Chinese learning investment as possible from documents (e.g., policy documents and media reports), the literature, and student interviews (conducted in Chinese and English) in order to build the Q set. When the interview data reached saturation, i.e., the point at which no new information emerged during the coding process, the total number of international students interviewed was 31 [40]. Participants in Q sorting were chosen from these students on the basis of gender, nationality, and funding type, for the purpose of enriching the findings. Most of them were third-year undergraduates who were not of Chinese origin (except for GU and YIN). The Q methodology is explained in the next section.

*3.3. Data Collection and Analysis*

The Q methodology, introduced in the field of psychology by William Stephenson, is a mixed method employed to explore issues related to subjectivity through a quantitative analytic process. Typically, a Q study consists of four basic steps: (1) generation of a Q set, i.e., a concourse usually including 40 to 80 statements; (2) "Q sorting", i.e., gathering viewpoints from the participants; (3) statistical analysis of the Q sorts; and (4) interpretation of the emergent factors [39,41]. It has gained popularity among researchers in the social sciences, including second language acquisition, as it helps to reveal the role of subjective feelings in language learning, which are relatively under-explored and increasingly catching researchers' attention [42]. As a methodology of data collection, it can be practically utilized in combination with a wide range of theoretical perspectives [27,43].

During the first step, we collected as many statements pertaining to investment in Chinese learning as possible, since a Q set is supposed to be broadly representative [41]. A total of 57 statements were collected from the literature and student interviews and finally pruned to 30 by removing similar or duplicate statements. In step 2, 15 (N = 1/3–1/2 of the statement number in the Q set) international students were recruited to constitute the P set, namely the participants in Q sorting whose opinions were gathered for further

interpretation. It should be noted that four additional participants (AI, YIN, MEN, and WA) were also invited to conduct the Q sorting in case of possible withdrawal of any of the Q sorters, but one of them did not finish the Q sorting and the other three did not generate eligible Q sorting data, so ultimately their Q sorting data were not included in the subsequent factor analysis. To enrich the findings, participants were chosen based on their diverse backgrounds (see Table 1). They were asked to rank the 30 statements in a fixed quasi-normal distribution (see Figure 1) with scores ranging from +3 ('most agreeable') to −3 ('most disagreeable'), based on the extent to which they agreed or disagreed with each statement. Finally, 15 eligible Q sorts were generated (see Table 2), and follow-up interviews were also conducted with the corresponding sorters to complement the Q sorting data. In step 3, we used PQ Method, a software program designed for Q analysis, to perform a statistical analysis of the Q sorting data. It should be noted that factor analysis in Q methodology differs from the traditional R methodology in that the former is characterized by a *by-person* correlation and factor analytic procedure while the latter typically employs *by-variable* factor analysis [39,41,44]. In other words, Q methodology focuses on the perspectives, viewpoints, or attitudes of a group of persons, rather than on the representativeness of participants. In the final step, we obtained some emergent factors, each representing a certain grouping of individuals who share similar perspectives and attitudes. Each factor was explicated in a way that resembles the process of coding in typical qualitative studies.

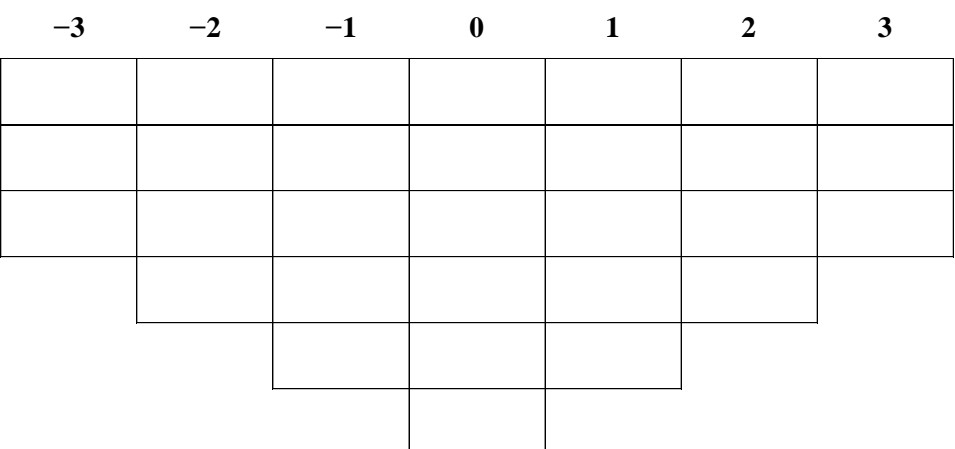

**Figure 1.** Fixed quasi-normal distribution used for the 30-card sorts.

**Table 2.** Factor matrix.

| Participants | Factor 1 | Factor 2 | Factor 3 |
|---|---|---|---|
| ME | 0.7842X | 0.0331 | 0.1946 |
| HU | 0.6148X | 0.251 | 0.1791 |
| BE | 0.0197 | −0.1409 | 0.7202X |
| BO | 0.0008 | 0.7169X | 0.1951 |
| LI | −0.339 | 0.5537X | −0.4617 |
| JI | 0.0392 | 0.5958X | −0.1517 |
| GU | 0.2912 | 0.5987X | 0.081 |
| MU | 0.2365 | 0.3438 | 0.5860X |
| NA | 0.7263X | 0.2543 | 0.3447 |
| JIN | 0.1811 | 0.5063X | −0.0136 |
| YA | 0.7710X | 0.0798 | −0.3035 |
| HE | −0.0901 | 0.0913 | 0.7098X |
| RU | 0.2233 | 0.6098X | 0.2539 |
| YI | −0.1248 | 0.7228X | −0.0039 |
| WE | 0.6429X | −0.0833 | −0.1228 |

Note: 'X' indicates a high correlation (i.e., a value ≥ 0.47, which is seen to be statistically significant) between a participant's Q sort and the factor; e.g., ME correlates 0.7842 with Factor 1.

## 4. Findings

Three factors emerged in the statistical analysis, each representing three types of incentives for Chinese learning investment, namely cultural capital and multilingual posture-oriented, economic capital-oriented, and cultural capital and experience-oriented incentives. Our qualitative interpretation of these three types of incentives is presented in this section.

### 4.1. Factor 1: Multilingual Posture and Cultural Capital-Oriented Investment

Five students were included in Factor 1: ME, HU, NA, YA, and WE. The statements with which they most agreed and disagreed are shown in Table 3.

**Table 3.** Statements most agreed and disagreed with by participants included in Factor 1.

| | Most Agreed:<br>I Learn Chinese . . . | | Most Disagreed:<br>I Learn Chinese . . . |
|---|---|---|---|
| +3 | Statement 20: in order to be a multilingual.<br>Statement 22: in order to enrich my life experience.<br>Statement 6: in order to satisfy my interest in Chinese culture (e.g., Kung Fu, Tai Chi, calligraphy). | −3 | Statement 8: in order to get a degree easily.<br>Statement 2: in the hope of getting the scholarship provided by China.<br>Statement 16: in order to be recognized by the parents/teachers/peers/boss of mine. |
| +2 | Statement 5: in order to satisfy my interest in the values and customs of China.<br>Statement 1: in the hope that I can find a job with high pay.<br>Statement 21: in order to be a person with cross-cultural awareness.<br>Statement 26: in the hope of having a chance to go abroad to get the experience of studying, communicating and contacting with students from various countries together. | −2 | Statement 13: in the hope that I can find a job respected by others.<br>Statement 27: in the hope of studying in a university away from my hometown and parents' discipline.<br>Statement 12: in the hope of having a chance to live in China for a long period in the future.<br>Statement 11: in the hope of being a member of a community in which people use Chinese to communicate. |

Table 3 revealed that students included in Factor 1 were, on the one hand, motivated by their multilingual (and multicultural) posture, e.g., to be multilingual (Statement 20) with cross-cultural awareness (Statement 21), and anticipating interaction with students from other countries (Statement 26). On the other hand, they also tended to be interested in Chinese traditional culture, e.g., Kung Fu, Tai Chi, and calligraphy, and appreciate Chinese values and cultural customs; in this sense, their investment in Chinese learning was mostly in exchange for specific cultural capital. However, they did not value academic qualifications or scholarships, recognition and respect from others, or the opportunity to integrate into Chinese-speaking society. Although finding a well-paid job (Statement 1) is also one of their most agreed-with statements, some statements pertaining to economic capital fell into the most disagreed-with category, e.g., getting the scholarship (Statement 2) and finding a job respected by others (Statement 13).

Take ME as an example. She came from Bangladesh and could speak Bengali, English, Hindi, and Urdu. She became interested in and started to learn Tai Chi during her MBA studies before coming to China, which marked the beginning of her investment in Chinese learning:

*In 2016, China's president went to Bangladesh. At that time, I began to study Tai Chi, but I didn't know it was Tai Chi. I thought it was Kung Fu . . . I loved Kung Fu and went to study it every Friday. I had learned it for two months. When China's president went there, the Confucius Institute arranged some performances for him. Then the teacher asked me – he had given me a Chinese name, xxx – if I would like to perform, since an important man would come. I said yes . . . I became the group leader... We practised for two months and performed for the president. My photo was published on the newspaper. I was very happy. (Interview, ME)*

ME's investment in Chinese learning was motivated by her interest in Chinese culture, specifically her experience of learning Tai Chi and performing for China's president. This falls into the category of embodied cultural capital, i.e., something that becomes part of the long-term disposition of an individual [21]. In addition, her investment was also driven by her desire to seek more employment opportunities (given the limited employment choices for women in her country) and for the experience of meeting people from other countries.

*In ancient Bangladesh, girls were not allowed to study; they had to stay at home and learned what their mothers did. It has been a tradition that the boys are thought to be smart, stronger, and they can be shown to others, while girls are not like that. This kind of belief is influential. I think girls who get good grades and don't want to give up are really admirable. (...) In a company nowadays, there are probably only two or three female executives in the highest positions, more men in the middle ranks, and more women in the lower ranks. Women in the highest positions are really, really exceptional. They attained powerful positions which are not easily taken away by others. ( . . . ) In my first year I wanted to go back to my country because few people spoke Chinese there and I could get a good job. Now I want to stay in China because I want to do a lot of things, but I don't have much freedom in my own country because my mum and dad won't let me do it; they want to protect me, they say one salary is enough, you don't need to... I don't like that, I want to do a lot of things, even if I fail. (Interview, ME)*

*The first day I came to the hotel, I saw my room-mate. She embraced me, and I thought [I found] someone who could know me. At the time of enrolment, I saw many foreigners. I began to think I was very lucky to see so many people from different countries here. I had much interest. I wanted to know and talked to them. There were 18 people in our class. They were from 15 countries. (Interview, ME)*

Like ME, students such as HU, NA, and WE were primarily motivated to learn Chinese partly to help find employment. However, their investments proved to be dynamic and complicated in the course of their Chinese learning. NA, for instance, remembered how her Chinese learning was first encouraged by a Chinese teacher, who told her that becoming a Chinese language interpreter in Columbia was a good prospect. While learning Chinese, she found her interest in Chinese characters growing, which further motivated her to invest in learning Chinese. In a sense, their multilingual posture may also be fostered by their need to enlarge their repertoire and discover more possibilities for self-expression, as in the example of YA:

*Now sometimes I find it hard to express what I think and feel. [In my daily life] we usually talk about things we buy from the shop or something like that, these are very simple things. In Chinese there are a lot of words about feelings, more specific words, and I want to learn these. (Interview, YA)*

It should be noted that although YA also claimed that her Chinese learning investment was partly motivated by her desire to be multilingual, she clarified that such a desire was not formed on an instrumental basis, i.e., it was not a desire to obtain economic benefits such as employment opportunities through building a Chinese-speaking social network. In other words, her Chinese learning investment was not driven by the pursuit of social capital, but instead by the benefits she perceived which pertained to emotional expression.

Overall, the multilingual posture and cultural capital-oriented incentives for investment felt by ME, HU, NA, YA, and WE were co-influenced by identity (self-positioning) and ideology [19]. They were not restricted by the fixed identity perception of being a person of a specific country; rather, they thought they belonged to a global society or were amidst cosmopolitans and locals. Such perceptions meant they believed they could learn more about other cultures, enrich their life experiences, find more ways to express themselves, and enjoy making friends with people from other cultures through their investment in Chinese learning. Additionally, their economic capital-oriented investment incentives were formed by the prevailing ideology that Chinese serves as an important business language and is becoming a lingua franca.

### 4.2. Factor 2: Economic Capital-Oriented Investment

Seven students were included in Factor 2: BO, LI, JI, GU, JIN, RU, and YI. Table 4 presents the statements with which they most agreed or disagreed.

**Table 4.** Statements most agreed and disagreed with by participants included in Factor 2.

| Most Agreed:<br>I Learn Chinese . . . | | Most Disagreed:<br>I Learn Chinese . . . | |
| --- | --- | --- | --- |
| +3 | Statement 1: in the hope that I can find a job with high pay.<br>Statement 23: in the hope that I can find a job I am interested in.<br>Statement 8: in order to get a degree easily. | −3 | Statement 19: in order to form the ability of identifying a specific things or express feelings more concretely.<br>Statement 16: in order to be recognized by the parents/teachers/peers/boss of mine.<br>Statement 27: in the hope of studying in a university away from my hometown and parents' discipline. |
| +2 | Statement 2: in the hope of getting the scholarship provided by China.<br>Statement 22: in order to enrich my life experience.<br>Statement 26: in the hope of having a chance to go abroad to get the experience of studying, communicating and contacting with students from various countries together.<br>Statement 25: to have a high-quality learning experience in China (including advanced teaching methods, interesting learning activities, etc.). | −2 | Statement 30: in order to take responsibility of my own choice of major.<br>Statement 29: in the hope of having a pleasant experience of communicating with others in a new language.<br>Statement 24: in the hope of living a higher quality life in China compared with life in my country.<br>Statement 6: in order to satisfy my interest in Chinese culture (e.g., Kung Fu, Tai Chi, calligraphy). |

Participants included in Factor 2 were characterized by economic incentives, e.g., scholarships or employment opportunities (in business, translation, or teaching), and appreciated the high-quality learning experiences available in China. It is noteworthy that the two most agreed-with statements were employment-related, while cultural capital and experiences, such as recognition from others, were not dominant elements in Factor 2.

*I didn't think too much at that time. I thought why I shouldn't go [to China] now that they gave me scholarship? Just have a try. I would go back if it didn't work. Without the scholarship, you should pay the money yourself or something like that. I didn't know whether I would continue to study or whether I liked it or not. (Interview, BO)*

BO invested in Chinese learning to gain a scholarship, then to get a bachelor's degree, and finally to find a high-paying job; this is a process of transforming cultural capital (educational qualifications) into economic capital [19].

Compared with BO, GU, and LI, others such as YI, JIN, JI, and RU also intended to gain economic capital, but with a clearer purpose, namely, to become an educator (e.g., a professor/Korean teacher/English teacher). JIN, for example, mentioned the prospect of teaching Chinese in Republic of Korea:

| | |
| --- | --- |
| *Researcher:* | *How is the situation of learning Chinese in your country nowadays?* |
| *JIN:* | *It's very popular, very, very popular. Some students from the Republic of Korea also pass the test of HSK 6, even though they haven't come to China. There are so many people like that, but that's the case in Seoul, and in other places . . . But still many parents send their children to tutoring centres for learning Chinese. They all do that for finding employment, since many companies require HSK certificates. Because of the development of China's economy and power, learning Chinese serves as an opportunity. (Interview, JIN)* |

Intriguingly, their pursuit of economic capital through learning Chinese seemed to be influenced by their imagined identities. YI, for example, dreams of becoming a professor in

the future, while GU hopes to become a businessperson. Their perceptions of their career prospects are part of their identity, the projection of their imagined identities in the future that shapes their investment at present [18]—YI and GU strive to realize their imagined identities through constant investment in Chinese learning.

In sum, students in Factor 2 were mainly motivated by economic capital, i.e., government scholarships or Chinese-related employment opportunities. This can be explained by the learners' imagined identities and the language ideology related to Chinese as a result of the increasing value of Chinese accompanied by the restructuring of economic strength and the development of China [22].

*4.3. Factor 3: Cultural Capital and Experience-Oriented Investment*

Three students were included in Factor 3: BE, MU, and HE. The statements with which they most agreed or disagreed are presented in Table 5.

**Table 5.** Statements most agreed and disagreed with by participants included in Factor 3.

| Most Agreed: I Learn Chinese . . . | | Most Disagreed: I Learn Chinese . . . | |
| --- | --- | --- | --- |
| +3 | Statement 6: in order to satisfy my interest in Chinese culture (e.g., Kung Fu, Tai Chi, calligraphy, etc.). Statement 2: in the hope of getting the scholarship provided by China. Statement 18: in order to satisfy my curiosity in something new. | −3 | Statement 24: in the hope of living a higher quality life in China compared with life in my country. Statement 1: in the hope that I can find a job with high pay. Statement 19: in order to have the ability to identify specific things or express feelings more concretely. |
| +2 | Statement 17: in order to experience the new challenge in life. ("Challenge" refers to learning Chinese.) Statement 29: in the hope of having a pleasant experience of communicating with others in a new language. Statement 26: in the hope of having a chance to go abroad to get the experience of studying, communicating and contacting with students from various countries together. Statement 20: in order to be a multilingual. | −2 | Statement 8: in order to get a degree easily. Statement 25: to have a high-quality learning experience in China (including advanced teaching methods, interesting learning activities, etc.). Statement 15: in order to get a sense of accomplishment in study. Statement 27: in the hope of studying in a university away from my hometown and parents' discipline. |

As is presented in Table 5, participants included in Factor 3 chose to invest in Chinese learning not only for their culture-related interests, but also for the enjoyable experience of Chinese learning itself. Take MU as an example:

*At first, I found some Chinese characters on my toys. I copied the Chinese characters and took them to school just to be cool. I used to pretend I could speak Chinese, but this became an awkward experience when I met some Asian people one day and my classmates asked me to communicate with them. Since then I have really started to learn Chinese by myself. I studied whatever materials about Chinese I found online at that time, such as some textbooks in PDF versions and Chinese-teaching videos on YouTube. I spent a year learning Pinyin. (Interview, MU)*

*When I was in Morocco I had only a few close friends around me and sometimes we didn't share the same preferences. After coming to China, I was surprised that I had more and more friends around me just because I could speak Chinese, and I sometimes wondered why that was, and it gave me quite a lot of motivation. (Interview, MU)*

For MU, being perceived as able to speak Chinese made him feel "cool" in his early years, and this sustained his investment in Chinese learning ever since. His investment in Chinese learning also brought him more friends with shared interests, which can be seen

as a kind of self-fulfillment, namely, being a member in his imagined community (friends united by shared interests) via Chinese learning investment.

Overall, the investment in Chinese learning by BE, MU, and HE was driven by their pursuit of Chinese culture-related benefits as well as their learning experience (which may enable them to bridge the gap between reality and their imagined identities).

## 5. Conclusions

Drawing on a combination of Q methodology, in-depth interviews, and the theoretical model of investment [22,39], this study has explored the investment of CSL students in their learning in terms of the benefits they perceive, which actually serve as incentives. The findings reveal that:

(1) The investment of CSL students is intra-personally and inter-personally diverse and can be divided into three categories: multilingual posture and cultural capital-oriented investment, economic capital-oriented investment, and cultural capital and experience-oriented investment. This indicates that their investment in learning Chinese is collaboratively driven by utilitarian incentives (e.g., scholarships, employment opportunities, and educational qualifications) which are interwoven with non-utilitarian incentives (e.g., the enrichment of one's experiences, expressing oneself, and making friends).

(2) CSL students' perceptions of the benefits of their investment not only involve different forms (economic, cultural, or social) of capital, but also some facets of their imagined identities (e.g., an international/multilingual/multicultural posture, or the desire for emotional expression), which are not necessarily convertible into capital.

Compared with the existing research on second language learning investment, this study has a more empirical basis and examines the way in which CSL learners' investment is shaped by their multi-faceted imagined identities. Moreover, interwoven utilitarian and non-utilitarian incentives emerged from the data, indicating that affective objectives also underlie learners' investment [45]. The utilitarian tendency was observed in most cases in this study. Specifically, economic capital-oriented investment (Factor 2) has a typically utilitarian basis, both at the individual and group levels, while the perceived benefits of the emotional or affective aspects of learning Chinese (which can be seen in Factor 1 and Factor 3) were also identified. In other words, the benefits perceived by learners that motivate their investment are not necessarily resources that contain some social value in the linguistic market, but are more similar to the "leisure" or "consumption" motivations found in some previous studies, i.e., the aspiration to socialize with others or express oneself and self-identify with one's imagined community [46,47]. In this study, such an imagined community manifested as a multilingual and multicultural community, wherein learners self-identify as multilingual and cosmopolitan, which leads to self-fulfillment rather than a mere "good return".

It should also be noted that though data from some of the Q sorters (i.e., AI, YIN, MEN, and WA) were not eligible to be included in the factor analysis, their interview data also revealed some intriguing perspectives. YIN, for instance, a girl of Chinese origin who can speak multiple languages (Portuguese, English, French, and Chinese), regards herself as an individual with a "bi-identity" due to her intercultural life experiences. YIN's perception of her "bi-identity" reveals her uncertainty about her cultural identity, while her self-identification as an "in-between" person provides room for her bi-/multicultural posture and inclination to self-identify as a bicultural person, although this does not necessarily lead to stronger investment in Chinese learning [26]. This makes it different from "transcultural identity", as the latter reflects the exercise of agency in facilitating language teaching/learning by deploying multilingual and multicultural resources [30].

Furthermore, the role of identity and ideology can be further understood if we scrutinize the way learners cope with their imagined identities as they relate to language learning. By constructing imagined identities (e.g., cosmopolitan, businessperson, or professor) for themselves, they can identify the perceived benefits pertaining to their target language(s), and are thus motivated to enhance their investment in learning. Additionally, in contrast to

the opinion of Darvin and Norton [22] that "outsiders" (in terms of race, ethnicity, gender, social class, etc.) may be marginalized in the target community, international students in China from different countries and with different cultural backgrounds use Chinese to communicate with each other and break the limits of any definite standard of a "legitimate speaker"; they embrace the ideological view that Chinese is a language with high economic value and that it is difficult but charming, and thus identify the potential return of learning Chinese.

Some limitations of this study must be addressed. Firstly, only one university was involved; comparative studies could be conducted in a greater number of educational institutions with diverse backgrounds. Secondly, longitudinal studies could be conducted to better capture the dynamics of investment in learning Chinese. Thirdly, CSL education policy texts should be taken into consideration in order to develop a macro-cum-micro understanding of CSL education planning.

**Author Contributions:** Conceptualization, J.L., Y.W., Q.S. and X.G.; original draft preparation, Y.W. and J.L.; supervision and review, X.G. and Q.S. All authors have read and agreed to the published version of the manuscript.

**Funding:** This research received no external funding.

**Institutional Review Board Statement:** The study was approved by the Research Ethics Committee of the Institute of Linguistics, Shanghai International Studies University.

**Informed Consent Statement:** Informed consent was obtained from all subjects involved in the study.

**Data Availability Statement:** The data presented in this study are available on request from the corresponding author. To protect the privacy of the research participants, the data are not publicly available.

**Acknowledgments:** Our sincere gratitude goes to all the participants involved in our study.

**Conflicts of Interest:** The authors declare no conflict of interest.

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
