# Peer review of "Investment in Learning Chinese by International Students Studying Chinese as a Second Language (CSL)"

_sustainability, doi:10.3390/su142315664_

Round 1
Reviewer 1 Report
Dear authors
what did you mean when in Abstract you repeated "economic capital oriented" twice (line 15)?
The same problem has been noticed in lines 206-207, 338-339 (We consider that the authors intended to speak about social investment as well, however they use economic investment rather than social)
In Tables 2, 3 and 4 we would recommend to make some amendments: instead of repeating "I learn Chinese" each time and with each statement, the authors would rather have this "I learn Chinese" on the summit of the table and then, the statement would continue "... in the hope that I ..."
Author Response
- What did you mean when in Abstract you repeated "economic capital oriented" twice (line 15)?
We are sorry for the ambiguous expressions here. We intend to show that participants loading on Factor 1 and Factor 3 were motivated by more than one incentive simultaneously, i.e. in the latest revised version, participants loading on Factor 1 were motivated by multilingual posture as well as cultural capital simultaneously, and participants loading on Factor 3 by cultural capital and experience simultaneously.
- The same problem has been noticed in lines 206-207, 338-339 (We consider that the authors intended to speak about social investment as well, however they use economic investment rather than social)
Apologies again. As is explained above, modifications of Factors 1 and 3 have been made and relevant expressions have been changed accordingly.
- In Tables 2, 3 and 4 we would recommend to make some amendments: instead of repeating "I learn Chinese" each time and with each statement, the authors would rather have this "I learn Chinese" on the summit of the table and then, the statement would continue "... in the hope that I ..."
Many thanks for the suggestion. We have reorganized Tables 3, 4 and 5 (i.e. most-agreed and most-disagreed statements in Factors 1-3) as suggested.

Reviewer 2 Report
Evaluation:
The current study examined international students’ motivation towards investing in Chinese learning. An inductive method, i.e. Q methodology, was exploited to investigate international students’ incentives for their Chinese language investment. The findings show that types of motivation on Chinese learning are highly relevant to economic capital, cultural capital or in between the cultural and economic capital. As there has existed a world-wide rise in learning Chinese as a second language, this study is a welcome addition to literature, which may bring about an understanding of learners’ subjectivity in Chinese language learning.
My comments concern ways to make the manuscript more elaborated in the sections of Data Analysis and Findings.
1. Overall, I have great concerns about the part of quantitative analysis in the current study. On page 4, lines 171-172, the authors mentioned that Q methodology “is a mixed method employed to explore issues related to subjectivity through a quantitative analytic process”. According to the authors’ further statements on page 4, line 174, the quantitative analysis seems to be present in Step 3 of this Q study, i.e., “(3) statistical analysis of the Q sorts”. Then, in the follow-up content, readers will highly expect to read more detailed information about how participants’ subjective Q sorts are statistically analyzed.
However, after reading the subsequent description on page 5, lines 194-198, I found that there was limited information about the process of the quantitative analysis. The authors expressed that factor analysis featuring by-person correlation was used to analyze Q-sorts, but there was a lack of statistical numbers or correlation tables as supportive content to demonstrate how it ended up with 3 identified factors in this study; what statistical numbers were used as the indication to show each participant’s correlation with different identified Factors.
I highly recommend a reference to the article of Ellingsen et al. (2010). Their study shows a clear example of how to present the statistical results of the factor loadings by participant and factor type. Table 1 in Ellingsen et al.’s study provides a correlation matrix, which provides a straightforward overview of the statistical results. The correlation coefficients in the matrix clearly demonstrates the way of statistically grouping participants in that participants who correlated with one factor have similar common viewpoints that differentiates them from those who correlated with the other factors. Would the authors provide a factor matrix of their statistical results?
Ellingsen, Ingunn & Størksen, Ingunn & Stephens, Paul. (2010). Q methodology in social work research. International Journal of Social Research Methodology. 13. 396-409. 10.1080/13645570903368286.
2. On page 5, lines 192-193, “15 eligible Q sorts were generated …”. Originally, there were 19 international students recruited for Q-sorting. Therefore, I’m concerned about what standards were applied to recognizing the 15 Q-sorts as eligible. Please explicitly provide the reasons why the 15 Q-sorts were kept, and the other 4 Q-sorts were eliminated.
3. In the section of Findings, on page 6, lines 214-215, it is stated that “…students loading on Factor 1 hoped to become multilingual, multicultural persons and enrich their life experiences…”. However, information displayed in Table 2 seems opposite to what is stated in the text. Table 2 shows that the statements of “I learn Chinese in order to satisfy my interest in the values and customs of China” and “I learn Chinese in order to be multilinguals” are found in the column of “Most Disagreed”. Concerning the first statement, “values and customs of China” to some extent refers to Chinese culture, since culture can be defined as customs, beliefs, values and symbols that a group of people accept or uphold. In other words, Table 2 implies that participants loading on Factor 1 are interested in neither Chinese culture nor being multilingual.
Therefore, my concerns are as follows: why participants loading on Factor 1 were classified as “cultural and economic capital-oriented investment”? The levels of agreement and disagreement among statements seem to suggest that Factor 1 can be identified as “economic capital-oriented investment”. Can the authors explain why the text content is inconsistent with what is displayed in Table 2?
4. Similarly, on page 7 line 266, I am confused about why the authors identified Factor 2 as “economic capital oriented”, when in Table 3, the statement “I learn Chinese in order to satisfy my interest in the values and customs of China” was found in the column of “Most agreed”. This statement to a great extent shows support to the cultural-oriented incentive of Chinese learning investment. Why did the authors interpret Factor 2 as purely economic capital oriented?
Minor issues:
5. On page 5, line 211, it is reported that “Five students were included in Factor 1”. However, the subsequent content only includes four participants’ assumed names: ME, HU, NA, and WE. It seems that one name is absent here. Please check this.
6. On page 9, line 300, Please add Factor 3 before “Multilingual and Multicultural Posture Oriented” to keep consistent in naming the section titles.
7. On page 11, line 365, please pay attention to the font size of the word “scrutinize”.
Author Response
- Overall, I have great concerns about the part of quantitative analysis in the current study. On page 4, lines 171-172, the authors mentioned that Q methodology “is a mixed method employed to explore issues related to subjectivity through a quantitative analytic process”. According to the authors’ further statements on page 4, line 174, the quantitative analysis seems to be present in Step 3 of this Q study, i.e., “(3) statistical analysis of the Q sorts”. Then, in the follow-up content, readers will highly expect to read more detailed information about how participants’ subjective Q sorts are statistically analyzed.
However, after reading the subsequent description on page 5, lines 194-198, I found that there was limited information about the process of the quantitative analysis. The authors expressed that factor analysis featuring by-person correlation was used to analyze Q-sorts, but there was a lack of statistical numbers or correlation tables as supportive content to demonstrate how it ended up with 3 identified factors in this study; what statistical numbers were used as the indication to show each participant’s correlation with different identified Factors.
I highly recommend a reference to the article of Ellingsen et al. (2010). Their study shows a clear example of how to present the statistical results of the factor loadings by participant and factor type. Table 1 in Ellingsen et al.’s study provides a correlation matrix, which provides a straightforward overview of the statistical results. The correlation coefficients in the matrix clearly demonstrates the way of statistically grouping participants in that participants who correlated with one factor have similar common viewpoints that differentiates them from those who correlated with the other factors. Would the authors provide a factor matrix of their statistical results?
Ellingsen, Ingunn & Størksen, Ingunn & Stephens, Paul. (2010). Q methodology in social work research. International Journal of Social Research Methodology. 13. 396-409. 10.1080/13645570903368286.
Many thanks for the constructive comments and suggestions. We have cited the recommended reference and insert the Factor matrix (i.e. Table 2 on Pages 5-6) as required, which not only shows the Q-sorters’ correlation with Factors 1-3 respectively, but also annotates with which particular factor each Q-sorter highly correlates (i.e. the absolute value of correlation ≥ 0.47).
Table 2. Factor matrix.
|
Participants |
Factor 1 |
Factor 2 |
Factor 3 |
|
ME |
0.7842X |
0.0331 |
0.1946 |
|
HU |
0.6148X |
0.251 |
0.1791 |
|
BE |
0.0197 |
-0.1409 |
0.7202X |
|
BO |
0.0008 |
0.7169X |
0.1951 |
|
LI |
-0.339 |
0.5537X |
-0.4617 |
|
JI |
0.0392 |
0.5958X |
-0.1517 |
|
GU |
0.2912 |
0.5987X |
0.081 |
|
MU |
0.2365 |
0.3438 |
0.5860X |
|
NA |
0.7263X |
0.2543 |
0.3447 |
|
JIN |
0.1811 |
0.5063X |
-0.0136 |
|
YA |
0.7710X |
0.0798 |
-0.3035 |
|
HE |
-0.0901 |
0.0913 |
0.7098X |
|
RU |
0.2233 |
0.6098X |
0.2539 |
|
YI |
-0.1248 |
0.7228X |
-0.0039 |
|
WE |
0.6429X |
-0.0833 |
-0.1228 |
|
Note: ‘X’ indicates a high correlation (i.e. the value ≥ 0.47, which is seen to be statistically significant) between a participant’s Q sort and the factor; e.g. ME correlates 0.7842 with Factor 1. |
|||
- On page 5, lines 192-193, “15 eligible Q sorts were generated …”. Originally, there were 19 international students recruited for Q-sorting. Therefore, I’m concerned about what standards were applied to recognizing the 15 Q-sorts as eligible. Please explicitly provide the reasons why the 15 Q-sorts were kept, and the other 4 Q-sorts were eliminated.
Thank you for pointing out the problem. At first there were 19 students invited to participate in the Q-sorting, while only 15 of them provided eligible data. Also, the number of Q-sorts was determined to be 15 since it falls into the range between 1/3 to 1/2 of the statement number (30) in the Q set. We have added the explanation in the section of Data Collection and Analysis.
- In the section of Findings, on page 6, lines 214-215, it is stated that “…students loading on Factor 1 hoped to become multilingual, multicultural persons and enrich their life experiences…”. However, information displayed in Table 2 seems opposite to what is stated in the text. Table 2 shows that the statements of “I learn Chinese in order to satisfy my interest in the values and customs of China” and “I learn Chinese in order to be multilinguals” are found in the column of “Most Disagreed”. Concerning the first statement, “values and customs of China” to some extent refers to Chinese culture, since culture can be defined as customs, beliefs, values and symbols that a group of people accept or uphold. In other words, Table 2 implies that participants loading on Factor 1 are interested in neither Chinese culture nor being multilingual.
Therefore, my concerns are as follows: why participants loading on Factor 1 were classified as “cultural and economic capital-oriented investment”? The levels of agreement and disagreement among statements seem to suggest that Factor 1 can be identified as “economic capital-oriented investment”. Can the authors explain why the text content is inconsistent with what is displayed in Table 2?
Many thanks for the correction. We apologize for the mistake on the serial numbers of the statements. Each statement and its serial number have been checked and modified in the revised manuscript. In the meantime, the label of each factor and data presented in the Findings section have also been revised accordingly.
- Similarly, on page 7 line 266, I am confused about why the authors identified Factor 2 as “economic capital oriented”, when in Table 3, the statement “I learn Chinese in order to satisfy my interest in the values and customs of China” was found in the column of “Most agreed”. This statement to a great extent shows support to the cultural-oriented incentive of Chinese learning investment. Why did the authors interpret Factor 2 as purely economic capital oriented?
Our apologies for this mistake again and modifications have been made as required.
- On page 5, line 211, it is reported that “Five students were included in Factor 1”. However, the subsequent content only includes four participants’ assumed names: ME, HU, NA, and WE. It seems that one name is absent here. Please check this.
Thank you for identifying the problem. The five students included in Factor 1 should be ME, HU, NA, YA, and WE. We have also made the correction in the manuscript.
- On page 9, line 300, Please add Factor 3 before “Multilingual and Multicultural Posture Oriented” to keep consistent in naming the section titles.
Many thanks for the suggestion. We have modified the title of Section 4.3 accordingly.
- On page 11, line 365, please pay attention to the font size of the word “scrutinize”.
Thanks a lot for the suggestion. We have changed the font size as required.

Round 2
Reviewer 2 Report
I appreciate the thorough and thoughtful response of the authors. They have sufficiently addressed my concerns with the earlier version of the manuscript.